# Photoprotection and Skin Pigmentation: Melanin-Related Molecules and Some Other New Agents Obtained from Natural Sources

**DOI:** 10.3390/molecules25071537

**Published:** 2020-03-27

**Authors:** Francisco Solano

**Affiliations:** Department of Biochemistry and Molecular Biology B and Immunology, School of Medicine and LAIB-IMIB, University of Murcia, 30100 Murcia, Spain; psolano@um.es

**Keywords:** photoprotection, sunscreen, melanin, antioxidant natural products, flavonoids

## Abstract

Direct sun exposure is one of the most aggressive factors for human skin. Sun radiation contains a range of the electromagnetic spectrum including UV light. In addition to the stratospheric ozone layer filtering the most harmful UVC, human skin contains a photoprotective pigment called melanin to protect from UVB, UVA, and blue visible light. This pigment is a redox UV-absorbing agent and functions as a shield to prevent direct UV action on the DNA of epidermal cells. In addition, melanin indirectly scavenges reactive oxygenated species (ROS) formed during the UV-inducing oxidative stress on the skin. The amounts of melanin in the skin depend on the phototype. In most phenotypes, endogenous melanin is not enough for full protection, especially in the summertime. Thus, photoprotective molecules should be added to commercial sunscreens. These molecules should show UV-absorbing capacity to complement the intrinsic photoprotection of the cutaneous natural pigment. This review deals with (a) the use of exogenous melanin or melanin-related compounds to mimic endogenous melanin and (b) the use of a number of natural compounds from plants and marine organisms that can act as UV filters and ROS scavengers. These agents have antioxidant properties, but this feature usually is associated to skin-lightening action. In contrast, good photoprotectors would be able to enhance natural cutaneous pigmentation. This review examines flavonoids, one of the main groups of these agents, as well as new promising compounds with other chemical structures recently obtained from marine organisms.

## 1. Solar Radiation and Skin Photodamage: General Concepts

Skin is an important barrier to protect the human body from environmental stress. One of the more important factors causing this stress is sun exposure, due to the energy and free radical generating capacity of sunlight. The solar radiation on the surface of our planet comprises an ample range of electromagnetic radiation [1], including ultraviolet (UV, approximate range of wavelength from 180 to 380 nm), visible (Vis, approximately from 380 to 800 nm), and infrared light (range 1–3 µm approximately). Among the wavelengths reaching the Earth’s surface, UV radiation is the most energetic and potentially harmful (Figure 1A).

UV radiation constitutes about 10% of the total energetic output of the sun. The biologically active composition of the UV radiation reaching the Earth has suffered some changes in the last decades due to chemical contamination and atmospheric factors [1]. This radiation is usually subdivided into three regions (UVA, UVB, and UVC). UVA comprises the longest wavelengths (320–380 nm), partially overlapping with the accompanying visible light, while UVB wavelengths are in the middle span (280–320 nm) and UVC comprises the shortest wavelengths (180–280 nm) with the highest energy. Fortunately, UVC rays do not penetrate through the stratosphere since the ozone layer acts as an efficient filter to deter the very harmful effects of such radiation (Figure 1A).

UVB, UVA, and visible light are partially filtered by the atmosphere, but the percentage of solar radiation arriving at the Earth’s surface is significant. In human skin, the penetrability is not uniform, and the most energetic UVB displays low penetrance due to the variety of cellular biochromes (mainly cutaneous pigments, proteins and nucleic acid) at the epidermis absorbing in this range of energy. In this way, the effects of UVB are mainly constrained to the epidermis. UVA and visible light show deeper penetrance, and they can affect cellular and extracellular structures in the dermis (Figure 1B). Certainly, UVA, as well as the most energetic region of visible light, blue light, are also important threats for skin aging, dryness, and carcinogenic transformation in the dermal layer.

It is obvious that solar radiation causes a number of beneficial effects on life. In addition to its essential role in photosynthesis in plants and microbial algae, Vis radiation is essential for the sense of sight in animals and regulation of circadian rhythms. UV light also has some important beneficial effects on human skin, including antibacterial capacity, stimulation of wound healing, jaundice prevention, and formation of active vitamin D from sterol precursors by the UV-induced opening of ring B [2,3]. However, long-term exposure to UV rays is a potential risk for skin damage, namely accelerated skin aging, such as wrinkling and sagging, sunburns, and even mutations leading to the promotion of different types of skin cancer.

To simplify the complexity of the mechanisms involved in these effects, UV light damages DNA by two different ways. The first one is produced through direct DNA absorption of mutagenic radiations, thereby increasing lesions called the UV “signature” (pyrimidine dimers or 6-4 photoproducts). The second mechanism is indirect, through interaction with other biochromes generating reactive oxygen species (ROS) which produce harmful cellular effects and can reach the nucleus causing oxidative DNA modifications and strand breaks [4,5,6]. In both cases, accumulative damage of DNA can finally induce apoptosis or lead to cancer appearance [5,7].

## 2. Natural Skin Pigments for Photoprotection: Melanin and Melanogenesis

Skin tone is related to the presence of several biochromes that contribute to the defense against solar radiation. Although some colored biomolecules such as hemoproteins or carotenoids contribute to skin tone, the most important pigment determining skin color is melanin [8].

Melanin is produced in specialized cells called melanocytes that are mostly distributed in the epidermal–dermal junction, and then distributed to surrounding keratinocytes, which are the most abundant cells in the epidermis. Melanocytes look like dendritic cells and contain specialized lysosome-lineage organelles called melanosome to synthesize and store the melanin. Melanosomes are transferred from melanocytes to the neighboring keratinocytes through elongated dendrites [9]. The approximate ratio among the population of melanocytes and keratinocytes in the basal layer of the human skin to supply melanin is around 1:30, although this number shows slight variations for different races. One melanin-forming melanocyte surrounded by keratinocytes and some other cell types is called the melano-epidermal unit [10]. In the cytosol of keratinocytes, melanosomes form a critical barrier/shield of DNA by forming perinuclear caps to exhibit photoprotection [11]. The physiological response of skin against solar radiation depends on the production, distribution, type, and quantity of melanin synthesized in melanocytes and transferred to keratinocytes (Figure 2).

Melanogenesis is the biochemical pathway leading to the synthesis of melanin [12]. This pathway (Figure 3) is initiated from the amino acid L-tyrosine, and the key enzyme is tyrosinase. Tyrosinase catalyzes two consecutive reactions, the hydroxylation of L-tyrosine to L-dopa and the concatenated oxidation of this o-diphenol to dopaquinone. Dopaquinone spontaneously cyclizes to form cyclodopa (also called leukodopachrome). These two intermediates (dopaquinone and cyclodopa) rapidly undergo a redox disproportionation to dopa (uncycled reduced product from dopaquinone) and dopachrome (cycled oxidized product from cyclodopa). Dopa is recruited to the pathway by the action of tyrosinase [13] and dopachrome continues the route for the formation of dark/brown eumelanin. Dopachrome suffers a slow spontaneous decarboxylation to 5,6-dihydroxyindole (DHI) or alternatively the enzyme dopachrome tautomerase (Tyrp2/DCT) catalyzes the formation of 5,6-dihydroxyindole-2-carboxylic acid (DHICA). DHI and DHICA are indolic o-diphenols easily oxidized either by tyrosinase or tyrosinase related protein 1 (Tyrp1) to the corresponding o-quinones. These species finally polymerize to form eumelanin. In addition to the enzymatic activity, TRP1 is also a stabilizing protein for tyrosinase, as its role is still unclear depending on the metal cofactor found at the active site [14,15]. DHI as well DHICA can be oxidized in the absence of any enzyme to their respective o-quinones by oxygen and ROS generated during previous reactions [16]. In any case, eumelanin is basically a polymer with mixture of decarboxylated (DHI) and carboxylated (DHICA) 5,6-oxygenated indole units at different oxidation degrees (5,6-dihydroxy, semiquinone and 5,6-quinone units).

On the other hand, dopaquinone can be conjugated with thiol-containing compounds, such as free L-cysteine or glutathione, to branch the pathway to derive sulfur-containing products leading to pheomelanin (Figure 3). Accordingly, human skin contains two types of melanin: eumelanin and pheomelanin. Their ratio determines the race and the Fitzpatrick skin phototype [17]. Eumelanin is dark, from black to brown depending of the DHI/DHICA ratio [18], whereas pheomelanin is a red or yellow pigment. Pheomelanin is predominant in light phenotypes, blond or red hair. Eumelanin is much more photoprotective than pheomelanin. In fact, after UV exposure, pheomelanin can easily become a photosensitized agent by stimulating lipid peroxidation and other reactions leading to a high amount of ROS and subsequent undesirable reactions [19,20].

## 3. Sunscreens: Parameters for Evaluating Photoprotection

It is clear that the photoprotection against solar radiation is due to dark skin pigmentation associated to eumelanin. In spite of the presence of a certain amount of melanin in human skin (aside from albinism), that amount is not enough in the case of light phototypes that preferentially contain pheomelanin. Thus, complementary sunscreens are essential for photoprotection. Sun exposure without skin protection can be harmful anytime and anywhere, particularly during the summertime. Sunscreens are used worldwide as an integral part of the photoprotection strategy [7,21,22,23,24]. It is assumed that chemical shielding over the skin provides protection against damage by all types of UV; although as already suspected, protection against sunburning may not directly correlate with skin cancer protection [25]. While this important point is remains unsolved, its discussion is beyond the scope of this review.

The photoprotective efficiency of sunscreens is determined through two main parameters: the sun protection factor (SPF) and the protection grade of UVA (PA). The standard parameter in the industry of sunscreens is the SPF, which directly measures protection against UV-induced skin erythema (sunburning) under standardized conditions. This effect is mostly due to UVB. According to U.S. Food and Drug Administration (FDA) regulations [26], commercial products must be labeled with SPF values that indicate how long they will protect against UVB radiation. SPF is generally in the range of 10–25, 25–50, and 50–100 (in fact 50+ according to the FDA), corresponding respectively to low, high, and very high protection [27]. People with phototype I need very high protection, a minimal SPF of 50. This is also the case for people suffering from diseases such as albinism and vitiligo, who are highly susceptible to the effects of UV due to failed melanin production. In constrast, dark phototypes, V and VI, would require lower protection, around 20.

The meaning of SPF should be well understood. Importantly, the SPF values mean protection capacity mostly against UVB light, but this is not sufficient in assessing the total amount of UV radiation entering the skin. On one hand, a SPF 20 sunscreen (protection 20) means that it absorbs 95% of the UVB radiations, while an SPF 50 product can block 98% of that radiation. Some arguments suggest that there is not much difference and both products supply a relatively high protection, 95% and 98%; but in terms of price market, a sunscreen with SPF 50 is significantly more expensive than one with SPF 20. However, when applying a SPF 50 product compared to a SPF 20 product, less than half of the erythematous UVB radiation (exactly only 40%) will penetrate into the skin, and this could become a big difference. On the other hand, standard SPF measurements have focused primarily on UVB wavelengths. Sunscreens should protect not only against UVB effects, but also against other risky regions (e.g., UVA, blue light, and even infrared). It should be taken into account that UVA shows deeper penetrance into human skin, and this radiation reaches the dermis and the epidermis–dermis junction where most melanocytes are sited. Hence, the effect of UVA on induction of pigmentation and possible alteration of melanocyte phenotype without erythema evidence due to UVB filtering could be significant yet. Unfortunately, this fact has been frequently ignored. Innovative sunscreens with improvements in addition to SPF value are needed as broad photoprotective agents.

Therefore, around 20 years ago, the Japan cosmetics industry introduced an alternative method to evaluate UVA efficacy of sunscreen [28] based on the results of in vivo persistent pigment darkening (PPD) on skin treated with a putative photoprotective product. Thus, in addition to SPF, the sunscreens could be labeled from PA+ to PA++++, corresponding to the level of protection grade of UVA (PA) obtained from the PPD test [29]. Sunscreens labeled as PA+ mean low protection, whereas PA++++ represents products that provide a high sunscreen efficiency [23,30]. In agreement with the protection efficiency against UV radiation, PA+ sunscreens contain poorer composition and lower amounts of products than PA++++ sunscreens. These are composed of at least eight UVA filters and other agents added to minimize the damage based on ability to absorb, reflect, and scatter solar radiation.

In addition to UV–Vis absorbing agents, recent sunscreens display other properties, such as water resistance, photostability, hydrating agents, sticking lotion to avoid reiterative application, etc. These points are involved in the measurement of the so-called Biological Effective Protection Factors (BEPFs), that are calculated for a determined UV-mediated skin response through transmission measurements according to the Diffey method [31] using the relative action spectrum [32]. BEPFs can be considered as indicators focused on the number of times that a person protected with the agent can be exposed in comparison to an unprotected person (control). So far, its determination and use in the information provided in commercial sunscreens are rare.

## 4. Photoprotective and Antiaging Components in Sunscreens: Nature, Sources and Action

The classification of sunscreens according to composition in organic, inorganic, and systemic products has been extensively examined in a number of reviews. Readers interested in this classical classification are referred to recent reviews [23,24] in addition to the large number of original papers related to particular products or molecules, including this Special Issue. 

This review is restricted to natural photoprotectors under two points of view. On one hand are photoprotectors based on melanin and melanin-like products, as this pigment is the main physiological pigment in human skin. On the other hand are photoprotectors based on some naturally pure products or alcoholic extracts obtained from natural sources, such as plants and marine organisms. In fact, this second approach has been used throughout the centuries in different ways, and many of these products come from ancient recipes for different purposes, but some are potentially good candidates to add to sunscreens. They can exert an anti-aging effect in addition to possible photoprotection by filtering or absorbing UV light, forming part of cosmeceutics. Modern sunscreens constitute a broad-spectrum group that is currently being incorporated into cosmeceutical products. Cosmeceutics can be widely defined as “topical formulations which were neither pure cosmetics, like lipstick, nor pure drugs, like corticosteroids” [33].

The mechanisms involved in appropriate photoprotection can be classified according to three non-exclusive processes:(i)Stimulation of natural protection through action on melanocytes and surrounding keratinocytes to stimulate the synthesis of endogenous melanin and its subsequent distribution through the melano-epidermal unit. Basically, they act by stimulating the tyrosinase activity and melanogenic proteins (Figure 3) and/or the subsequent transference of melanosomes to keratinocytes (Figure 2).(ii)Use of biopolymers related to natural melanin or unpolymerized molecules to cover the skin as an external shield against solar damage. Nevertheless, some of these biopolymers can partially penetrate inside epidermal keratinocytes.(iii)Stimulation of other mechanisms of the skin not directly related to pigmentation, but involved in retarding the aging, oxidizing, and inflammatory effects of sun exposure. Maintenance of hydration or collagen is somehow a way of photoprotection, minimizing accelerated wrinkling, sagging, and solar elastosis.

### 4.1. Natural Compounds Related to Animal Melanin

Concerning the use of natural melanin as a component in sunscreens, it is clear that all eumelanins show photoprotective properties due to their strong UV and visible light absorption and antioxidant properties [18,19,34,35]. Their absorption coefficient decreases at longer wavelengths, so they are especially appropriate for UV photoprotection.

Eumelanin is insoluble in water, although in combination with other molecules can become slightly water-soluble, and it forms emulsions with lipids easily. Thus, there are a number of proposals, publications, and direct patents, related to the diverse use of eumelanins as tanning agents or as possible components of sunscreens. They are found in a number of organisms. The source proposed for extraction is diverse, including sepia melanin [36], squid ink [37], green tea [38], *Pseudomonas maltophylia* [39], and chemical oxidation from dopa and other melanin precursors [40,41,42]. However, perhaps due to aesthetic reasons, taking into account the unpleasant dark color of any eumelanin-containing sunscreen, the use of all these patents has not been exploited in the commercial market. Nanocomposites of melanin nanoparticles from sepia ink reinforced with cellulose nanofibers have been recently prepared and characterized. This material improves stability and color appearance, but the melanin moiety keeps the UV blocking capacity and its antioxidant activity [43].

Alternatively, in the last years, other melanin-related compounds have been proposed to increase the absorption and general properties of natural melanin in order to be used in cosmeceutics. In this regard, in exploring the maintenance role of the carboxyl group in polymers, studies revealed that the use of DHICA methyl ester improves the antioxidant and protective effects of the final polymer, particularly by the higher absorption of UVA light [16,44].

A complementary approach has been the use of dopamine, the decarboxylated analogue of dopa. Dopamine can be polymerized easily to a polymer called polydopamine. The tailoring strategy of the polymerization of DHI in comparison to DHICA and DHI mixtures has been deeply studied [45]. This polymer shows several applications in nanobiotechnology [46]. The homopolymer of DHI is very similar to natural eumelanin (Figure 4), which is a polymer of DHI and DHICA as mentioned above [18], but simpler and probably larger in size as the only monomer unit is DHI [45,47].

Eumelanin is mainly formed by bonds between positions 4 and 7 of the indole ring, as the carboxyl group of DHICA blocks position 2 and deactivates conjugation to position 3. However, DHI is able to polymerize through positions 2, 3, 4, and 7 (Figure 4), giving place to a larger polymer able to absorb a large amount of UV light due to the large aromatic π-electron clouds [35]. In turn, polydopamine is free of protein and other compounds occurring in natural eumelanins extracted from living organisms. 

Under suitable conditions, polydopamine produces melanin-like nanoparticles resembling the melanin granules found in sepia ink. When a suspension of these nanoparticles is incubated with human keratinocytes, they are absorbed by the cells and distributed around their nuclei like natural melanin [11]. In that way, the material resembles natural melanosomes or melanin granules, and it functions as a pigment to immediately darken skin, but moreover, it also protects the DNA from UV light and can become more permanent than other components of sunscreens that are not absorbed through the epidermal cells. 

Related to that, the skin cancer incidence ratio between Caucasian and black skin is around 60, giving an approximate idea of the gap between different phenotypes that should be covered by the sunscreen requirements of both races. The cutaneous distribution of melanin for DNA protection is more abundant but also more concentrated in the basal layer of black skin than in its counterpart, Caucasian skin [48]. 

In addition to polydopamine, other polymer models mimicking natural melanin are being investigated as catechol/quinone heterodimers [49], but they clearly belong to the photoprotectors obtained by chemical synthesis rather than by natural means. 

### 4.2. Natural Products from Plants, Herbs, and Marine Organisms

There is ample evidence proving that photoprotectors play a critical role in inducing natural pigmentation as well as reducing the incidence of UV-induced human skin disorders such as melasma and hyperpigmented spots [22,50]. However, pigmentation is just an important aspect of photoprotection, but not the only one. UV radiation generates large amounts of ROS radicals in the skin. They stimulate melanogenesis by accelerating the reactions among dihydroxyindoles and indole quinones at the last phase of melanogenesis, increasing skin tanning. However, ROS also lead to the activation of other biological processes that are involved in UV-induced skin aging, wrinkling, and sagging [51]. In fact, ROS induce a number of cellular signaling pathways leading to inflammation and the acceleration of cellular senescence by damaging the structure of the dermal matrix, particularly collagen and proteoglycan degradation. Exposure to UV radiation activates the expression of aging-related genes through regulation of a variety of signal pathways such as Akt, GSK3β, and mitogen-activated protein kinases (MAPKs)-activator protein 1 (AP1) through inhibition of p38 phosphorylation [52,53]. 

According to the appearance of ROS, the use of antioxidants as adding agents of sunscreens is not surprising [54]. The application of antioxidants would result in prevention of these biological effects, as they scavenge ROS and restrict the extension of harmful chain reactions. The optimal antioxidants should have high UV-absorption but with good stability to avoid becoming photosensitizers after light absorption, be non-toxic, and show penetrance in the skin. The UV-absorption is related to π-electron systems, which are mainly sited as conjugated bonds in linear chain molecules or as aromatic compounds. These π-electron systems also confer ROS scavenging and antioxidant capacities. 

In agreement with that, there are a large number of agents with putative beneficial effects that can be putatively added to sunscreens. They include ascorbate [55], tocopherols [56], carotenoids [57], polyphenols [58], and flavonoids [59]. However, all antioxidants are not equivalent. Carotenoid action is based on absorption of UV light and quenching of singlet oxygen [60], polyphenols join light absorption with quenching ROS, and tocopherols are a family of antioxidant molecules especially effective in preventing cell membrane oxidation [61]. Antioxidants such as ascorbate do not have π-electrons and can be easily and rapidly oxidized by atmospheric oxygen, thus losing its efficacy as a photoprotector. 

Plant extracts with phenolic and flavonoid compounds are more effective [62,63]. The antioxidant capacity of natural extracts is not always proportional to the total content of tocopherols, carotenoids, polyphenols, or flavonoids [63]. This can be due to the diverse tests used (gallate or Trolox equivalent antioxidant capacity, ferric reducing antioxidant potential, and the 2,2′–azobis(2–methylpropionamidine)-induced oxidation of linoleic acid) or the different and complementary mechanisms of those families of compounds to exert their antioxidant activity. Biological tests using epidermal cell culture or cosmeceutical trials are necessary to provide a reliable estimation of the efficacy of natural extracts concerning antioxidant activity and efficacy as sunscreen components. Furthermore, a large number of plants, herbs, microalgae, and marine organisms are continuously tested for their putative potential in cosmeceutics. Some of them are discussed as follows.

#### 4.2.1. Plants and Herbs

Some of the natural sources containing these antioxidants are: (1) aloe vera (*Aloe barbadensis Miller*), known as a healing plant, it contains aloesin, anthraquinones, and saccharides [64], and is very effective in the amelioration of UV radiation-induced skin effects [65]; (2) tomato extracts (*Solanum lycopersicum L.*) contain lycopene, a carotenoid UV-absorbing molecule; (3) cactus extracts (*Opuntia humifusa*), that induce dermal hyaluronic acid production; [66]; (4) green tea (*Camellia sinensis*), which possesses anti-inflammatory activity due to the constituent epigallocatechin-3-gallate (EGCG) [67]; (5) pomegranate (*Punica granatum*) extracts, containing punicic acids and polyphenols that prevent dermal damage by inhibition of the UV-induced matrix metalloproteinases (MMPs) [68]; (6) alcoholic extract of *Ranunculus bulumei* (an Indonesian herb), recently demonstrated to be very effective, which attenuates the expression in UVB-irradiated HaCaT transformed keratinocytes of MMP9 and Cyclooxygenase-2 (COX-2) genes and enhances the expression of Type 1 collagen as well as Hyaluronan Synthase-2 (HAS-2) and HAS-3 [69], involved in skin hydration. COX-2 is an inducible enzyme involved in the formation of prostaglandin E2 from arachidonic acid and other eicosanoids.

However, it should be noted that the photoprotective effects of these agents are not clearly proven in all cases. The effect of adding lycopene to sunscreens is poor, as it has a modest or low increase in the SPF [70]. In other cases, such as the addition of cucumber extracts (*Cucumis sativus*), scientific data indicate that there is no correlation with any increase in the SPF value [71]. Other extracts, such as that obtained from the Indian beech tree (*Pongamia pinnata,* L.), just help in broadening the UV protection ability and avoiding the undesired effects of synthetic sunscreen compounds such as p-Amino Benzoic Acid (PABA) [72]. Moreover, the use of methoxypsoralens from bergamot oil (*Citrus bergamia*) after sun exposure increases photosensitivity, resulting in greater damage rather than photoprotection in spite of its stimulatory effect on tyrosinase activity [73].

Unfortunately, antioxidants are usually tyrosinase inhibitors. Inhibition of melanogenesis is a very common approach for hypopigmenting treatments [74]. The literature concerning skin pigmentation is full of reviews about the structure and mechanisms of tyrosinase inhibitors [50,75]. These compounds, usually found in natural plants and herbs, inhibit melanogenesis by reducing the expression of Microphthalmia-associated Transcription Factor (MITF), the master regulator of gene expression for the melanogenic system, through its effects on cAMP-Regulated Transcriptional Co-activator (CRTC) and cAMP response element-binding (CREB) activity, and subsequently the expression of tyrosinase and the tyrosinase-related protein genes. Thus, they are normally used as skin-lightening products in cosmeceutics. Among these compounds, one of the most studied and characterized is resveratrol obtained from grape vine (*Vitis vinifera*) [76,77] or some of its derivatives, such as α-viniferin, a trimer found in the same fruit and also in the Siberian pea tree (*Caragana sinica*) [78]. Those potent hypopigmenting agents protect skin keratinocytes from inflammation and oxidative damage rather than enhancing melanization.

However, ideal antioxidants for photoprotection should possess antiaging effects with the induction of melanogenesis. This is difficult from a chemical point of view, since melanogenesis is an oxidative process enhanced by the appearance of ROS in the final steps of the pathway (Figure 3), and antioxidants usually scavenge ROS. However, the compatibility between antioxidants and stimulation of melanogenesis is sometimes possible. In fact, there are some reports in the scientific literature in this field pointing out this fact, particularly regarding flavonoids [79]. The main groups of the thousands of flavonoids found in nature are depicted in Figure 5. The comparative analysis of flavonoids and their effects on melanogenesis to establish a correlation among chemical structure and melanogenic activity is complex. It was found that flavonols including quercetin, kaempferol or fisetin, flavones including luteolin, apigenin, and chrysin, and isoflavones including genistein showed stimulation of melanogenesis. However, rutin, robinetin, myricetin, epigallocatechin gallate, and naringin did not show any effect or they behave as inhibitors of melanin formation [79,80]. It seems that a hydroxyl group bound to ring B is a requirement for stimulation of melanogenesis (Table 1). It should be taken into account that some of these compounds are phenols and therefore alternative substrates of tyrosinase, giving place to the possibility of forming melanin-like dark oligomers. Thus, flavonoids with an o-diphenolic side seem to be the favorite structures to activate melanogenesis through tyrosinase activation.

However, the correlation among different flavonoid structures and melanin stimulation is not definitive yet. Flavonols activate melanogenesis, but rutin (a glycon form of quercetin), myricetin, and robinetin (both with three hydroxyl groups at ring B) do not, suggesting that the presence of a saccharide or three hydroxy groups on ring B prevent the effect (Table 1). Flavanols as the EGCG, with hydroxyl esterified with gallic acid, act as skin-whitening products rather than melanin activators. Surprisingly, flavones are activators of melanogenesis in spite of the absence of the hydroxyl group at position 3 of ring C, regardless of the number of hydroxyl groups on ring B (luteolin has two, apigenin one and chrysin none). Flavanones such as naringin are not activators either, probably due to the glycon moiety, since recently an aglycon flavanone such as sterubin was proposed to improve hair pigmentation as an activator of melanogenesis and melanocyte proliferation [81]. Finally and illustrating the difficulty in predicting the effect of flavonoids on melanogenesis, the apigenin-7-butylene glucoside, a glycon form of flavonoids, also enhances melanogenesis of B16–F10 cells by stimulation of the melanogenic proteins, tyrosinase and the associated, Tyrp1 and Tyrp2; as a result, this agent has been proposed for vitiligo treatment [82].

Data at Table 1 should be considered with caution. According to the o-diphenol nature of ring B, quercetin (3,3′,4′,5,7-pentahydroxylflavone) could be one of the most active flavonoids for promoting melanogenesis [80]. However, it must be also be mentioned that this is not completely accepted in cosmeceutics, and other reports indicate that quercetin possesses antioxidant and subsequently anti-melanogenic activities; it can be useful for skin whitening but is not effective as a photoprotector [83,84]. This discrepancy is not only found for quercetin, and further research should be carried out to clarify the actual effect of flavonoids on melanogenesis.

Recently, it has been reported that other flavonoid-like compounds and plant-derived products employed in traditional Asian medicine recipes also promote melanogenesis and are useful for vitiligo treatment. Chalcones, one class of flavonoid compounds, has become an interesting target for the development of anti-vitiligo agents. A series of novel chalcone derivatives have been synthesized and evaluated for biological activities [52]. Among them, derivative 1-(4-((3-phenylisoxazol-5-yl)methoxy)phenyl)-3-phenylprop-2-en-1-one (PMPP) was identified as a potent tyrosinase activator with higher action and lower toxicity than methoxypsoralen.

#### 4.2.2. Marine Organisms and Microalgae 

Marine organisms produce thousands of poorly known, but biologically-efficient molecules, and they have become one of the most promising sources of new molecules with antibacterial, antitumor, and anti-inflammatory activities with application in skin care. In the current context, these organisms also contain antioxidants and potentially photoprotective agents [85]. Many of these antioxidants have yet to be identified, but they are probably similar to those mentioned in the previous section. Others are new compounds with possible applications to this field. For instance, topsentin which is an alkaloid isolated from the marine sponge *Spongosorites genitrix* exhibits anti-inflammatory activity in UVB-irradiated human epidermal HaCaT keratinocytes. The photoprotective effect of topsentin was confirmed in a reconstructed human skin model [86]. Topsentin suppresses the expressions of two key targets for the prevention and treatment of skin damage and inflammation after sun exposure, COX-2 and miR-4485. COX-2 is an inflammatory mediator, as mentioned above, and miR-4485 is a nuclear microRNA that can be translocated to mitochondria affecting mitochondrial function in response to different stress conditions. This microRNA modulates the mitochondrial complex, production of ATP, caspase-3/7 activation, tumor necrosis factor (TNF)-α production, ROS levels, and apoptosis [87].

Microalgae also offer opportunities in the development of novel cosmeceutics focused on skin pigmentation [88]. Microalgae are found in a wide range of habitats including fresh and saltwater and marine environments, frequently associated with many marine invertebrates, including corals and sponges. Thus, Safafar et al. [89] reported that antioxidant compounds are abundant in a number of common microalgae, such as *Phaeodactylum* sp., *Nannochloropsis* sp., *Chlorella* sp., *Dunaliella* sp., and *Desmodesmus* sp. Characterization of the molecular nature of the active components is sometimes achieved, such as with polyphenols p-coumaric acid and apigenin [90]. However, the concentrations are low in comparison with the contents found in plants, but the beneficial effects of microalgae extracts are generally superior to the effect of just one of the components. 

Amongst the microalgal compounds that show inhibition of melanogenesis, carotenoids are prominent. Fucoxanthin has been reported to decrease tyrosinase activity in UVB-irradiated guinea pigs and the mRNA levels of melanogenic enzymes in UVB irradiated mice and skin cells [91]. Thus, most microalgae extracts are used as skin-lightening agents, such as a hydroalcoholic extract of *Chlamydomonas reinhardtii*, which inhibited melanogenesis in 3D human skin [92] or extracts of *Nannochloropsis gaditana* that showed a direct in vitro inhibition of tyrosinase [93].

However, it cannot be assumed that all microalgae extracts show skin-whitening activity because they are complex mixtures and the resulting final effect depends on the overall balance of many combined effects. Although the effects are never very pronounced, they contribute to protect the skin against sun exposure by a certain stabilization of tyrosinase activity against proteolytic degradation. These extracts would contain compounds such as some polyphenols, flavonoids, and saturated fatty acids, in addition to carotenoids [94,95]. 

Finally, there is growing interest for a family of structurally different products resulting in the secondary metabolism of marine organisms for use as photoprotectors and skin tanners. They are mycosporine-like amino acids, characterized by a cyclohexenone or cyclohexenimine chromophore conjugated with one or two amino acids. They are being proposed as ecologic photoprotectors. One of the most studied is palythine (Figure 5), extracted from corals and sea hares. These molecules show high photostability and strong UV-absorption, centered in the UVA-blue light range, from around 310–360 nm [96] that have evolved for protection against chronic sunlight exposure. Recent data indicate that very low concentrations of palythine show a significant protection to human keratinocytes submitted to UVA irradiation [97]. In addition, palythine behaves as a potent antioxidant, reducing oxidative stress by scavenging ROS species. These results suggest that they will have interesting potential as natural and biocompatible photoprotectors for adding to sunscreens independently of melanin pigmentation.

## 5. Conclusions

Sun radiation arriving at the Earth’s surface comprises an ample range of electromagnetic radiation, including UV light. This light is potentially harmful for human skin both by direct action on DNA or indirectly by generating ROS and oxidative stress. The effects range from just accelerating solar elastosis (wrinkling appearance and other skin aging effects), inflammation, and sunburns up to skin cancer promotion. There is no doubt that photoprotection to minimize these effects is essential. Human skin contains melanin, a quite efficient photoprotective pigment; but this is not enough, especially for light phenotypes. There is a necessity for sunscreens containing a variety of molecules to contribute to photoprotection. Any agent leading to stimulate melanogenesis would be helpful. Photoprotectors are very different in their structure, source, and mechanism of action, so sunscreens are very complex mixtures. This review has been dedicated to some of those molecules obtained from natural sources. Firstly, eumelanin, typically obtained from sepia ink. Other melanin-related molecules, especially polydopamine, were proposed. They are highly UV-absorbing polymers, and their SPF and PA are very high. However, so far, the commercial use of these substances is scarce, probably due to aesthetic problems derived from the color of the sunscreen containing those compounds. Secondly, other natural compounds obtained from plants, herbs, and marine organisms are employed. Basically, they should be UV-absorbing substances with ROS scavenging capacity and subsequent antioxidant activity. In terms of SPF and PA, none is as efficient as natural eumelanin. However, some of them gathered relative protection due to UV-absorption with the stimulation of natural melanogenesis. In this sort, flavonoids and some new marine molecules are being studied as they seem better than other natural antioxidants, such as vitamins C and D, carotenoids, or polyphenols. This review tries to collect and discuss data about some of these. It was concluded that further research is needed, as the balance between antioxidant activity with skin-lightening action and stimulation of melanogenesis is subtle and still partially unknown. In fact, the proposed photoprotective effect of commonly studied flavonoids, such as quercetin, is controversial as contrary data can be found in the scientific literature reporting the enhancement of melanin formation but also skin-lightening effects.

## Figures and Tables

**Figure 1 molecules-25-01537-f001:**
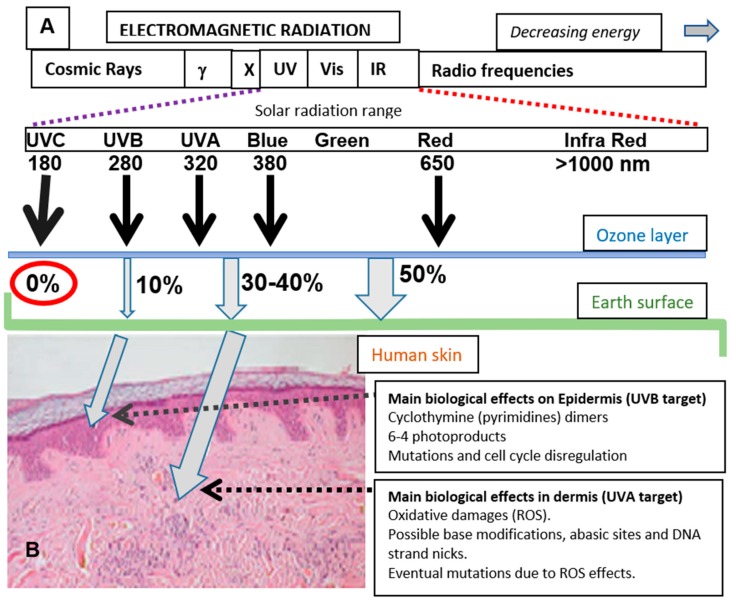
Solar radiation reaching Earth’s surface, skin penetrance, and biological effects. (**A**) Approximate percentage (%) of the total solar radiation reaching Earth’s surface for different wavelength regions. (**B**) Skin penetrance of UVB and UVA. Note that the less energetic UVA radiation has deeper penetrance than UVB. The main cellular consequences are also mentioned.

**Figure 2 molecules-25-01537-f002:**
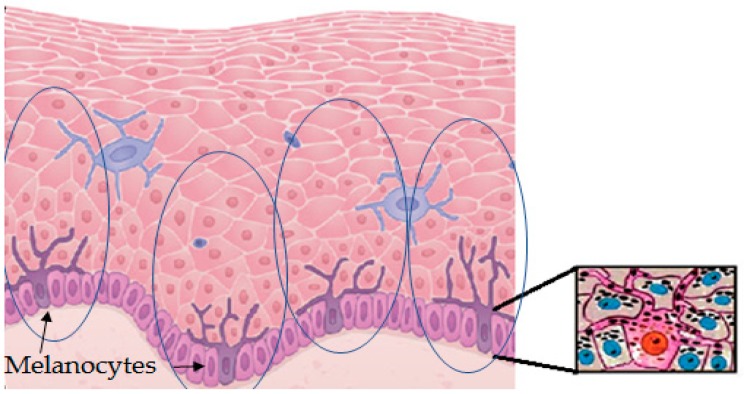
Graphic representation of a detail of the epidermis. It contains layers with abundant keratinocytes and some other cell types. Melanocytes are mostly found in the epidermal–dermal junction. Melanocytes are the only cells able to synthesize melanin. These cells have dendrites along the keratinocytes of their melano-epidermal units to facilitate the transport and transference of melanosomes filled with melanin. The image on the right shows one melanocyte transferring melanosomes (melanin granules, black dots) to surrounding keratinocytes of the melano-epidermal unit.

**Figure 3 molecules-25-01537-f003:**
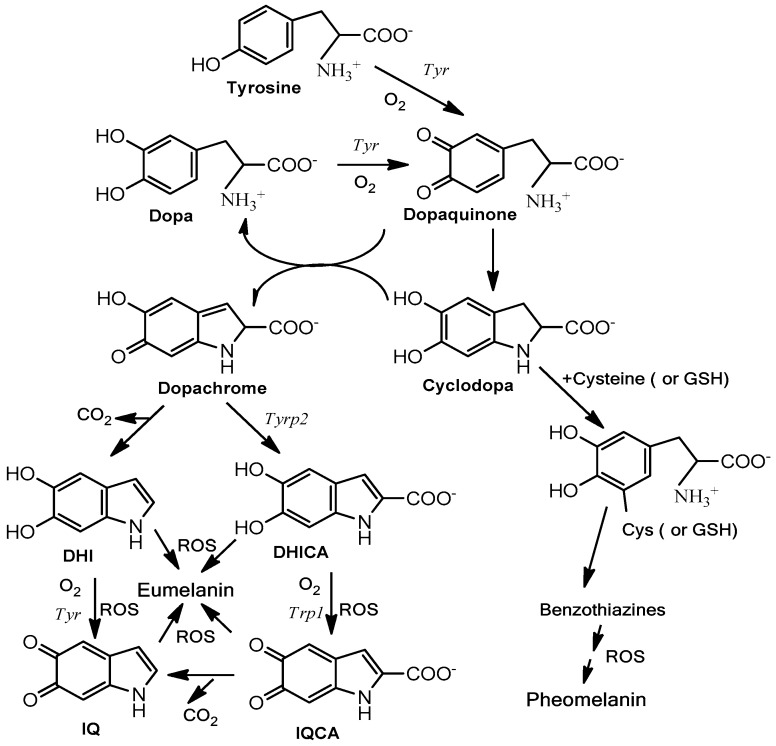
Schematic pathway of eumelanogenesis and pheomelanogenesis. Tyr is tyrosinase, the key enzyme catalyzing the rate-limiting step of the pathway, the oxidation of L-tyrosine to L-dopaquinone. Tyrp1 and Tyrp2 are involved in catalytic actions at the final phase of eumelanogenesis (left). Reactive oxygenated species accelerate the polymerization of indole units to eumelanin. Tyrps can also act as stabilizers of tyrosinase. On the other hand, dopaquinone is a pivotal branch point and can react with thiol-containing products, such as L–cysteine or glutathione (GSH), to lead the pathway to phaeomelanin through the intermediates cysteinyl-dopa and benzothiazine compounds (right). DHI, 5,6-dihydroxyindole; DHICA, 5,6-dihydroxyindole-2-carboxylic acid; IQ, 5,6-indolequinone; IQCA, indolequinone-2-carboxylic acid.

**Figure 4 molecules-25-01537-f004:**
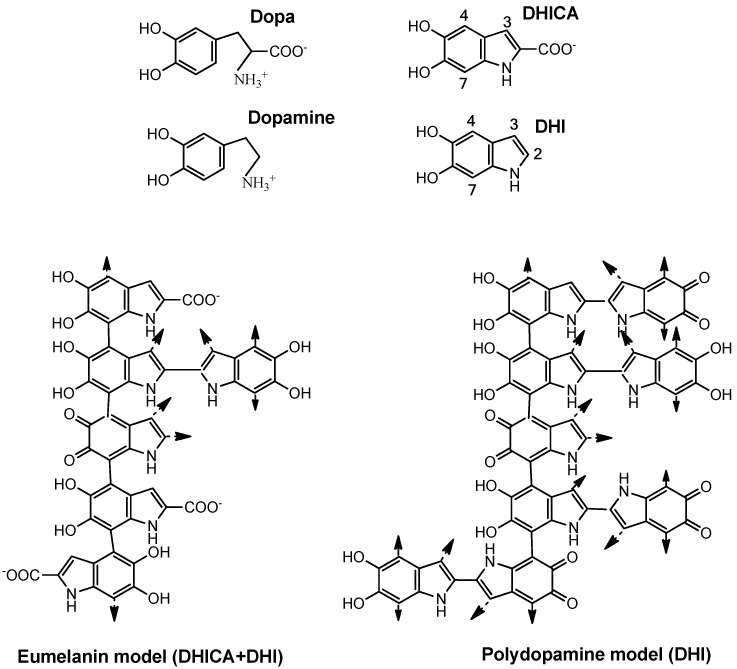
Structure of the precursor units (dopa, dopamine), the oxidized cyclized indole derivatives, DHI and DHICA, and a simplified model of eumelanin and polydopamine. Natural eumelanin might be considered a DHICA and DHI polymer. Polymerization occurs mainly through positions 4 and 7 since the presence of carboxyl groups blocks position 2 and greatly deactivates position 3. The size and light absorption of eumelanin depends on the DHI/DHICA ratio. The polydopamine model might be considered as a polymer of only DHI, more branched through positions 2, 3, 4, and 7. These are oversimplified models, since other uncycled units can be incorporated to the polymer during the uncontrolled formation of the pigments.

**Figure 5 molecules-25-01537-f005:**
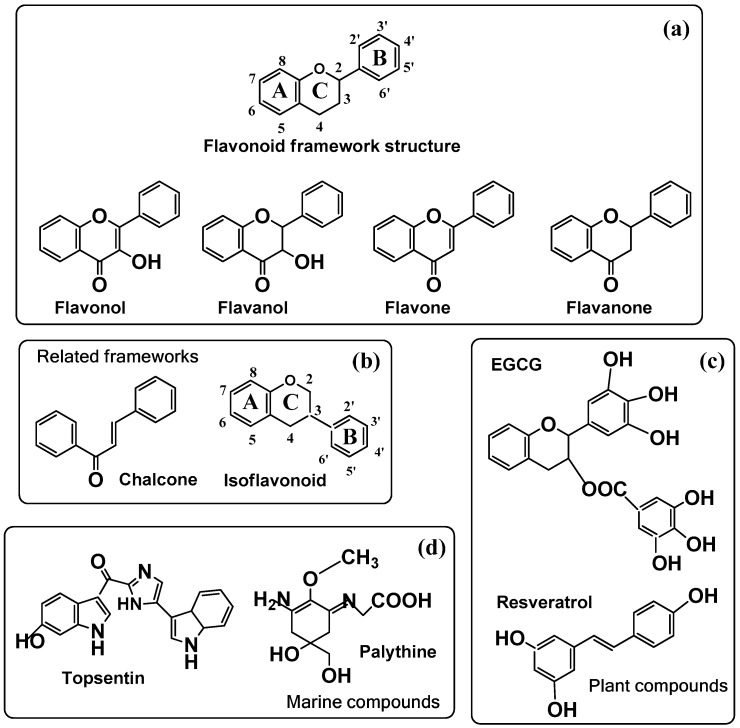
(**a**) General structure of the flavonoid framework with the four main types of flavonoids; (**b**) other flavonoid-related frameworks; (**c**) particular plant compounds and (**d**) marine compounds used in cosmeceutics (see Table 1). (**a**) The second line includes the four main types of flavonoids: flavonols, flavones, flavanones, and flavanols. (**b**) Flavonoid-related structure of the isoflavonoid framework (ring B is bound to position 3) and chalcone; (**c**) two important plant compounds, epigallocatechin-3-gallate (EGCG) found in green tea and resveratrol found in grape vine; (**d**) marine topsentin from a sponge and palythine from coral.

**Table 1 molecules-25-01537-t001:** Main flavonoids and related biomolecules used in cosmeceutics. Structure and melanogenic (+) or skin-lightening (−) activity [79,80,81].

Agent	Melanin action	Chemical name
**Flavonols**		
Galangin	−	3,5,7-trihydroxyflavone
Kaempferol	+	3,4′,5,7-tetrahydroxyflavone
Quercetin	+	3,3′,4′,5,7-pentahydroxyflavone
Rutin	−	3,3′(O-glycoside),4′,5,7-pentahydroxyflavone
Myricetin	−	3,3′,4′,5′,5,7- hexahydroxyflavone
Fisetin	+	3,3′,4′,7-tetrahydroxyflavone
Robinetin	−	3,3′,4′,5′,7-pentahydroxyflavone
**Flavones**		
Chrysin	+	5,7-dihydroxyflavone
Apigenin	+	4′,5,7-trihydroxyflavone
Apigenin-7-butylene Glycoside	+	4′,5,7-(O-butyl-glycoside)-trihydroxyflavone
Luteoin	+	3′,4′,5,7-tetrahydroxyflavone
**Flavanones**		
Narigin	−	4′,5,7(O-glycoside)-dihydroxyflavanone
Naringenin	−	4′,5,7-trihydroxyflavanone
Sterubin	+	3′,4′,5,7(methoxy)- tetrahydroxyflavanone
**Flavonoid-related molecules**		
EGCG	−	EpiGalloCatechin Gallate (actually a Flavanol)
Genestein (isoflavonoid)	+	4′,5,7- trihydroxyisoflavone
Chalcone	+	Benzyliden-acetophenone
Resveratrol	−	3,5,4′-trihydroxy-*trans*-stilbene
Topsentin	+	6-hydroxy-1H-indol-3-yl)-[5-(1H-indol-3-yl)-1H-imidazol-2-yl]-methanone
Palythine	+	2-[[3-amino-5-hydroxy-5-(hydroxymethyl)-2-methoxycyclohex-2-en-1-ylidene]-amino]-acetic acid

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
