# Peer review of "Photoprotection and Skin Pigmentation: Melanin-Related Molecules and Some Other New Agents Obtained from Natural Sources"

_molecules, 2020, doi:10.3390/molecules25071537_

Round 1

Reviewer 1 Report

Francisco Solano reviews the biological mechanism of photoprotection, the parameters often employed to evaluate the degree of photoprotection, and the natural compounds that have been employed as photoprotectors, while at the same time are able to enhance natural pigmentation.

The paper is well organized, is written in a very didactive way, and deals with a very interesting topic. I enjoyed very much reading the review. However, the English language should be revised since there many grammatically wrong and non-sense sentences and several typos. In addition, I suggest the following minor modifications. I recommend the publication of this work after addressing these minor revisions and improving the English language.

  1. The author should add a paragraph or section highlighting the main conclusions of the review.
  2. Figures 1 and 2 should have higher quality.
  3. Figure 3 is not too scientific. It would be more useful that this figure displays the chemical structure of the compounds involved in the different pathways.
  4. It would be more comfortable for the reader that Figure 4 labels positions 2, 3, 4 and 7 of DHI and DHICA.

Author Response

Francisco Solano reviews the biological mechanism of photoprotection, the parameters often employed to evaluate the degree of photoprotection, and the natural compounds that have been employed as photoprotectors, while at the same time are able to enhance natural pigmentation.

The paper is well organized, is written in a very didactive way, and deals with a very interesting topic. I enjoyed very much reading the review. However, the English language should be revised since there many grammatically wrong and non-sense sentences and several typos. In addition, I suggest the following minor modifications. I recommend the publication of this work after addressing these minor revisions and improving the English language.

Thank you very much for your right and smart comments- All of them are helpful for improving the manuscript. I would like to address all of them. I am glad about your allusion to the didactic aim of the review. I assume your comments about the English. The manuscript has been edited as well as modified according the suggestions of all the reviewers

  1. The author should add a paragraph or section highlighting the main conclusions of the review.

Right. A conclusion paragraph has been added

  1. Figures 1 and 2 should have higher quality.

I did my best, but I have serious limitations. I am confined at home since March, 12nd due to the devastating COVID-19. I have no facilities enough at home to cover all of them. 

  1. Figure 3 is not too scientific. It would be more useful that this figure displays the chemical structure of the compounds involved in the different pathways.

I did not include that structures in the previous version since the pathway is well-known and there are many publications containing that information, but according your request, the figure has been redrawn including the structure of the intermediates.

  1. It would be more comfortable for the reader that Figure 4 labels positions 2, 3, 4 and 7 of DHI and DHICA.

Ok, it is done. See DHI and DHICA units at Figure 4.

Reviewer 2 Report

Although several review articles have shown natural ingredients regulating skin pigmentation (PMID: 29552273), Solano et al well summarized in this manuscript the photoprotection and skin pigmentation with molecular and physiological mechanism as well as melanin-related compounds as photoprotectors.

There for this manuscript recommended for acceptance after minor revision.

Minor comments

  1. Word missing should be carefully confirmed. For example, several symbol was not able to read in page 8.
  2. The authors can provide reference number within Table 1

Author Response

Although several review articles have shown natural ingredients regulating skin pigmentation (PMID: 29552273), Solano et al well summarized in this manuscript the photoprotection and skin pigmentation with molecular and physiological mechanism as well as melanin-related compounds as photoprotectors.

Thank you very much for your right and smart comments- All of them are helpful for improving the manuscript. I would like to address all of them.  I agree with this general consideration. There are several excellent recent reviews about the subject. I did not include the review mentioned by the reviewer in the reference`s list since I tried to focused to photoprotection and that review is focused to management of hyperpigmentation, so that a number of skin-lightening products are not convenient as photoprotectors. I tried to write a review focusing to complementary aspects and I did not include the suggested review. Nevertheless, I took into consideration the reviewer´s suggestion and I have added that recent review in the bibliography (cited as ref. 50).   

There for this manuscript recommended for acceptance after minor revision. Minor comments

  1. Word missing should be carefully confirmed. For example, several symbol was not able to read in page 8.

I am sorry about that. Thank you. These symbols were Greek letters. All of them have been identified and corrected.

  1. The authors can provide reference number within Table 1

Done. The main references for Table 1 are no. 79, 80, 81, 86 and 97. They are discussed throughout the text, but they has been also added to the head of the Table.

Reviewer 3 Report

This paper is interesting and is potentially relevant to this journal. However, it needs further detail to make it appropriate for 'Molecules', and to allow readers to draw conclusions on the value of the compounds discussed. Examples of specific corrections are:-

No data is provided to demonstrate the effectiveness of the compounds and approaches discussed and this must be added throughout.

Very few structures of molecules are provided. I recognise that the authors state that in some cases the molecular entities responsible for the effects are not known, but in some cases they are known and have not been provided. They must be provided to make the review suitable for this specific journal rather than a different journal. 

No critical appraisal is given to allow the reader to decide which class is best and why. This must be added.

Although the focus of this review is on natural photoprotectors, the authors still need to put the work in context with other photoprotectors so that the reader can decide the value of these approaches.

There is no conclusion and this is a major weakness.

The written style needs improving especially for the non-introductory section.

The abstract needs rewriting as it is not clear what the aims of the review were, and it is not clear what the main findings are, and how these sit in the context of other approaches.

Author Response

Comments and Suggestions for Authors

This paper is interesting and is potentially relevant to this journal. However, it needs further detail to make it appropriate for 'Molecules', and to allow readers to draw conclusions on the value of the compounds discussed. Examples of specific corrections are:

No data is provided to demonstrate the effectiveness of the compounds and approaches discussed and this must be added throughout.

Thank you very much for your right and smart comments. I should admit that the reviewer is right and most of the comment might be pertinent, but this point is difficult to address as this is not the aim of the review.  There is no doubt that the effectiveness of dark eumelanin is higher than other natural compounds, but I am not involved in measurements of SPF and PA of different compounds, and furthermore, there is very few if any comparative data about this topic in the scientific literature. I have included a conclusion and some sentences throughout the manuscript about this concern.

Very few structures of molecules are provided. I recognise that the authors state that in some cases the molecular entities responsible for the effects are not known, but in some cases they are known and have not been provided. They must be provided to make the review suitable for this specific journal rather than a different journal. 

About chemical structures, figure 3 has been supplemented with the structure of the intermediates, Figure 4 shows the structure of a model for melanin and for polydopamine, and Table 1 and Figure 5 contain all information for drawing the structure of all flavonoids. Particular important compounds, both from plants or marine organisms, are provided (see c and d boxes). Concerning the Journal chosen for this submission, this is submitted to a special issue entitled “Natural and Artificial Photoprotective Agents”. I received an invitation for possible contribution, I proposed a tentative title and then I prepared this review. You are right that under other circumstances, other Journals could be also possible.  

No critical appraisal is given to allow the reader to decide which class is best and why. This must be added.

Melanin should be the most effective one in terms of photoprotection capacity, but they have an aesthetic problem for cutaneous application. Flavonoids are a good alternative, since they absorb UV light, and they are antioxidants but at the same time, some of them seem to stimulate endogenous melanogenesis. Marine products, such as mycosporin-like amino acids, are also good candidates to be used in sunscreens. These points are treated in the review, but unfortunately, I cannot say which one is best and why. There is no a unique answer to that. A lot of factors are involved. 

Although the focus of this review is on natural photoprotectors, the authors still need to put the work in context with other photoprotectors so that the reader can decide the value of these approaches.

Thank you again for this suggestion, but this one is similar to previous ones and therefore difficult to address in this review. There are some citations to non-natural protectors, such as polydopamine, but the sort and variety of photoprotectors is very wide and it can be found in the literature. As stated at lines 191-194, the classification of sunscreens according the composition in organic, inorganic and systemic products has been extensively treated in other reviews. Take into account that the review is focused to photoprotection of natural products rather than advanced chemical studies of melanin-related polymer. The claim of the reviewer might be pertinent in an ample perspective, but this review does not answer it. I take into considerations the suggestions for future thoughts.

There is no conclusion and this is a major weakness.

Right. A conclusion paragraph has been added. This conclusion covers some of the previous criticisms

The written style needs improving especially for the non-introductory section.

This is also right. I am sorry for that. I performed and extended edition of the English and the style. I honestly think that the manuscript has been improved in both content and style.

The abstract needs rewriting as it is not clear what the aims of the review were, and it is not clear what the main findings are, and how these sit in the context of other approaches. 

The abstract has been rewritten and a conclusion has been added. I hope this satisfied, at least partially, the claims of the reviewer. I am afraid that the reviewer claims for contextualizing with other approaches would extend the review out of my scope, experience and current circumstances. I am confined at home cannot satisfied this request. Anyway, thank you very much for your criticisms. I learn about your ample and pertinent points of view.

Reviewer 4 Report

The review by F Solano presents the potential of natural compounds either related to the melanin pathway or from plant sources as photoprotective agents. An extensive presentation of issues related to skin photoprotection and sunscreens classification and evaluation is also provided.

The review covers a topic of current  investigation in the dermocosmetic field applications. The presentation of critical and still debated issues may be of interest to the readers of Molecules. I have some suggestions to improve the readability and presentation

  1. It seems that there too many, probably useless, headings levels: only one main heading (# 1) is used with up 4 sublevels
  2. Paragraph 1.4.1

lines 229-230: please comment and quote also some recent papers reporting use of melanins from indole precursors with peculiar absorption features in the UVA region (Pigment Cell Melanoma Res. 2018, 31:475-483. doi: 10.1111/pcmr.12689. Int J Mol Sci. 2018 19 pii: E1753. doi: 10.3390/ijms19061753)

lines 240-248: for an updated discussion on the structure of melanins based on DHI and DHICA components and their contribution to the absorption spectrum of melanin see also Sci. Rep. 2017, 7, 41532; Angew. Chem., Int. Ed. 2013, 52,12684-12687. As to polydopamine, the structural features are still a subject of lively debate but the presence of uncyclized units at substantial levels is now commonly accepted.  Acc. Chem. Res. 2014, 47, 3541-3550

The above considerations should also be reflected in the structures shown in figure 4

  1. Figure 5 I found this figure quite confusing. Why not represent also the general structure of chalcone and isoflavone inside the frame and leave outside only the structures of the specific compounds that are discussed  in the text (like EGCG resveratrol etc) .
  2. Also I would suggest to show the structures of the compounds of marine origin in a separate figure

A concluding paragraph could be useful and possibly also a schematic illustration of those structural features e.g. in flavonoids that have been associated to a significant melanin pathway activation and photoprotection  

Other points

  1. Line 58 omit “chemical” UV induced ring opening seems enough
  2. Fig 1 the top illustration panel A appears of poor quality and shows some words not in English (radiofrecuencies)
  3. Line 99 “dismutation” seems not to be appropriate to describe a redox exchange between two different species.
  4. Line 276-278 the meaning of this sentence is not fully clear. The oxidation of dihydroxyindoles to the corresponding o-quinones is brought about by ROS.
  5. Line 298 one the main photoprotective action of carotenoids is quenching of singlet oxygen (see for example Antioxidants 2019 Jul 11;8(7). pii: E219. doi: 10.3390/antiox8070219)
  6. Line 324 should be read “HaCaT and fibroblasts ??”

Minor points misspellings or misprints

  1. Abstract line 16
  2. Fig 1 panel B “ciclothymine” should be cyclobutane thymine dimers
  3. Line 47 “since”  better because of or thanks to …
  4. Line 106 forms should be plural
  5. Line 132 it is clear… “that”
  6. Line 137 “that”
  7. Line 360 genistein
  8. Figure 5 EGCG lacks two OH groups on ring A

Author Response

The review by F Solano presents the potential of natural compounds either related to the melanin pathway or from plant sources as photoprotective agents. An extensive presentation of issues related to skin photoprotection and sunscreens classification and evaluation is also provided.

The review covers a topic of current investigation in the dermocosmetic field applications. The presentation of critical and still debated issues may be of interest to the readers of Molecules. I have some suggestions to improve the readability and presentation

Thank you very much for your right and smart comments- All of them are helpful for improving the manuscript. I would like to address all of them. Particular replies follow.

  1. It seems that there too many, probably useless, headings levels: only one main heading (# 1) is used with up 4 sublevels.

One heading level has been omitted to avoid the 4th sublevel.

  1. Paragraph 1.4.1

lines 229-230: please comment and quote also some recent papers reporting use of melanins from indole precursors with peculiar absorption features in the UVA region (Pigment Cell Melanoma Res. 2018, 31:475-483. doi: 10.1111/pcmr.12689. Int J Mol Sci. 2018 19 pii: E1753. doi: 10.3390/ijms19061753).

Thank for this suggestion. This has been done it (lines 235-239), although these new references are cited in other related paragraphs

lines 240-248: for an updated discussion on the structure of melanins based on DHI and DHICA components and their contribution to the absorption spectrum of melanin see also Sci. Rep20177, 41532; Angew. Chem., Int. Ed. 201352,12684-12687. As to polydopamine, the structural features are still a subject of lively debate but the presence of uncyclized units at substantial levels is now commonly accepted.  Acc. Chem. Res. 201447, 3541-3550

The above considerations should also be reflected in the structures shown in figure 4

Thank you. First of all, I have to say that I am aware of the extent and excellent work of the group mentioned by the reviewer in all these references and many others. I perfectly agree with the reviewer that Figure 4 is an oversimplification. The structures are undoubtedly more complex, and the incorporation of uncycled units has been widely suggested and discussed by that group as well as others. These considerations have been briefly incorporated to the manuscript. However, the definitive structure is still largely unknown, and it depends on the conditions of polymerization. Under this general view, I am afraid that the figure 4 cannot illustrate all possibilities of polymerization, both for natural melanin and for polydopamine. I would prefer leaving it without any alteration. This review is focused in a general view of photoprotection based on natural agents, and it is not focused to an updated view of eumelanin and polydopamine structure. In the same way, it is not focused in the cellular signaling to activate melanogenesis after sun exposure. MSH, MITF and other very important molecules are also briefly treated. I have omitted some of my own references to minimize the extension and numbering of the (mini)review. Otherwise, the review would become very large. I hope that reviewer would agree with these limitations. Anyway, I would insist in thanking for all these pertinent comments. Of course, my recognition to the contributions of the reviewer. According to that, the suggested references have been incorporated to the revised version, See the manuscript for a detailed view of the modifications

  1. Figure 5 I found this figure quite confusing. Why not represent also the general structure of chalcone and isoflavone inside the frame and leave outside only the structures of the specific compounds that are discussed in the text (like EGCG resveratrol etc) .

Thanks for this new suggestion. Chalcones and isoflavones were not included on the general structure of the four main types of flavonoids as they are not proper flavonoids. On the other hand, there are too many flavonoids in Table 1 to show the structure, but two particular agents, EGCG and resveratrol, are quite important in cosmeceutics and they are not flavonoids. I decided to incorporate these two particular structures in order to look for clarity. According to the reviewer, I have reorganized the figure using different boxes, and the figure legend has been corrected. I hope this rearrangement contributes to clear up the figure.   

  1. Also I would suggest to show the structures of the compounds of marine origin in a separate figure

These 2 selected structures have been included separately, as box (d), to emphasize that they are not flavonoids nor structurally-related flavonoid compounds.

A concluding paragraph could be useful and possibly also a schematic illustration of those structural features e.g. in flavonoids that have been associated to a significant melanin pathway activation and photoprotection.

Table 1 contains a column concerning that. On the other hand, the text emphasizes that this is a still unsolved question, as there are opposite reports. See for instance data about quercetin (lines 393-399). This controversy have been incorporated to the added brief conclusion section.

 Other points

  1. Line 58 omit “chemical” UV induced ring opening seems enough

Ok, thanks. Done

  1. Fig 1 the top illustration panel A appears of poor quality and shows some words not in English (radiofrecuencies)

Ok, it should be radio frequencies. The word was misspelt. These are the electromagnetic wave frequencies between audio and infrared. About quality of the figure, I did my best in the current unbelievable and devastating circumstances. Unfortunately, I am confined at home and I cannot use University facilities.

  1. Line 99 “dismutation” seems not to be appropriate to describe a redox exchange between two different species.

I had observed the use of the term “dismutation” in other previous papers, so that I also used. Looking for alternatives, perhaps the term disproportionation is chemically more correct. Consequently, I have replaced dismutation by disproportionation.

  1. Line 276-278 the meaning of this sentence is not fully clear. The oxidation of dihydroxyindoles to the corresponding o-quinones is brought about by ROS.

Right. Indeed, the oxidation of dihydroxyindoles to o-quinones is accelerated by ROS, but the sentence tried to introduce that ROS can also trigger undesirable effect leading to skin aging. The sentence has been re-written. Thank you.

  1. Line 298 one the main photoprotective action of carotenoids is quenching of singlet oxygen (see for example Antioxidants 2019 Jul 11;8(7). pii: E219. doi: 10.3390/antiox8070219)

Ok, Thanks again for this consideration. This feature of carotenoids has been incorporated to the manuscript, as well as the provided reference (lines 301). However, other reports indicated that carotenoids are not especially good as photoprotectors. These was the reason of focusing the second part of this manuscript in flavonoids. Anyway, I assume the lively controversy about this topic.

  1. Line 324 should be read “HaCaT and fibroblasts ??”

Sorry. It was a mistake. HaCaT line are transformed keratinocytes. This has been repaired

Round 2

Reviewer 3 Report

The authors have addressed most of my concerns, or have provided justification as to why this is not possible. I therefore consider it to be suitable for publication. The written style needs improving in places but this can be done in house.